# Optimization of Biocompatibility for a Hydrophilic Biological Molecule Encapsulation System

**DOI:** 10.3390/molecules27051572

**Published:** 2022-02-27

**Authors:** Alyssa B. Sanders, Jacob T. Zangaro, Nakoa K. Webber, Ryan P. Calhoun, Elizabeth A. Richards, Samuel L. Ricci, Hannah M. Work, Daniel D. Yang, Kaitlyn R. Casey, Joseph C. Iovine, Gabriela Baker, Taylor V. Douglas, Sierra B. Dutko, Thomas J. Fasano, Sarah A. Lofland, Ashley A. Rajan, Mihaela A. Vasile, Benjamin R. Carone, Nathaniel V. Nucci

**Affiliations:** 1Department of Molecular and Cellular Biosciences, Rowan University, 201 Mullica Hill Rd., Glassboro, NJ 08028, USA; alyssa.bree.sanders@drexel.edu (A.B.S.); zangaroj3@rowan.edu (J.T.Z.); nakoa.kristen.webber@drexel.edu (N.K.W.); calhounr9@rowan.edu (R.P.C.); richardse1@rowan.edu (E.A.R.); yangd3@students.rowan.edu (D.D.Y.); caseyk33@students.rowan.edu (K.R.C.); bakerg5@students.rowan.edu (G.B.); tjfasano1@gmail.com (T.J.F.); rajana55@students.rowan.edu (A.A.R.); mihaela.vasile@iqvia.com (M.A.V.); carone@rowan.edu (B.R.C.); 2Department of Physics & Astronomy, Rowan University, 201 Mullica Hill Rd., Glassboro, NJ 08028, USA; samuel.ricci@nih.gov (S.L.R.); hannah.work@cuanschutz.edu (H.M.W.); joseph.2.iovine@uconn.edu (J.C.I.); douglast3@knights.ucf.edu (T.V.D.); sierradutko@gmail.com (S.B.D.); sal214@pitt.edu (S.A.L.)

**Keywords:** protein encapsulation, reverse micelle, viability, fluorescence spectroscopy

## Abstract

Despite considerable advances in recent years, challenges in delivery and storage of biological drugs persist and may delay or prohibit their clinical application. Though nanoparticle-based approaches for small molecule drug encapsulation are mature, encapsulation of proteins remains problematic due to destabilization of the protein. Reverse micelles composed of decylmonoacyl glycerol (10MAG) and lauryldimethylamino-N-oxide (LDAO) in low-viscosity alkanes have been shown to preserve the structure and stability of a wide range of biological macromolecules. Here, we present a first step on developing this system as a future platform for storage and delivery of biological drugs by replacing the non-biocompatible alkane solvent with solvents currently used in small molecule delivery systems. Using a novel screening approach, we performed a comprehensive evaluation of the 10MAG/LDAO system using two preparation methods across seven biocompatible solvents with analysis of toxicity and encapsulation efficiency for each solvent. By using an inexpensive hydrophilic small molecule to test a wide range of conditions, we identify optimal solvent properties for further development. We validate the predictions from this screen with preliminary protein encapsulation tests. The insight provided lays the foundation for further development of this system toward long-term room-temperature storage of biologics or toward water-in-oil-in-water biologic delivery systems.

## 1. Introduction

Biological drugs, especially proteins, have emerged as promising avenues for treatment of many diseases [1,2,3]. A primary limiting factor in the movement of protein-based drugs from the lab to the clinic is the need for delivery systems that preserve the native, functional state of the proteins and protect them from degradation [4,5,6]. These limitations cause most protein-based therapeutics to require intravenous or inhalant delivery methods. These routes, while effective, show limited bioavailability and often result in rapid metabolism by enzymes or negative response by the immune system [7]. Oral delivery of protein-based drugs, while clinically ideal, also presents a distinct set of problems including degradation by the digestive system and difficulty associated with hydrophilic drug molecules penetrating epithelial cells [8].

To counteract these limitations, an optimal vehicle for protein-based therapeutics should safeguard cargo from degradation, allow for tissue-targeted delivery, and preserve the native structure of the protein. Many studies have aimed to facilitate development of such a system via encapsulation of proteins using a variety of nanoparticle-based approaches. Nanoparticle-based small molecule drug delivery platforms have matured greatly in recent years and are now quite sophisticated [9]. Solid polymeric nanoparticles, or nanocapsules, offer the potential for both slowed and environmentally controlled release via modulation of the polymer cross-linking and use of pH-dependent polymers, respectively [6]. While these platforms work well for hydrophobic drugs, the matrices require charging to encapsulate hydrophilic drug compounds. This approach shows strong utility for hydrophilic small molecule drugs, but protein-based drugs generally denature and/or aggregate due to excess matrix charge [10]. Extracellular vesicles and liposomes are promising alternatives because they contain biocompatible surfactants, can encapsulate proteins with minimal structural disruption, and are a known physiological mechanism for biomolecule transport [11,12]. Vesicular drug delivery, however, faces challenges from poor drug loading and rapid clearance of these vesicles in vivo [6,9].

Hybrid encapsulation strategies aim to combine the advantages of matrix-based nanoparticles and lipid-based encapsulation methods while offsetting their respective limitations. Solid lipid nanoparticles (SLNPs), hydrogels, and water-in-oil-in-water emulsions (W/O/W) have all shown promise [13,14,15]. SLNPs are similar to solid, matrix-based nanoparticles, but are instead composed of biocompatible lipids that are solid at room and physiological temperatures. The SLNP advantages are two-fold: they have the potential to deliver both hydrophilic and hydrophobic drugs, and they provide slow-release mechanisms, regulated through nanoparticle degradation in vivo [13]. A serious limitation of SLNPs is that the encapsulation of hydrophilic drugs requires high temperatures that can degrade protein drugs [16]. Hydrogels can be prepared without high temperatures, can provide high encapsulation efficiency for hydrophilic drugs, and offer low toxicity when prepared from biocompatible matrices. Encapsulation of proteins in hydrogels requires either strong electrostatic protein–matrix interactions or the presence of surface-exposed amino acids that facilitate covalent linking to the hydrogel matrix [14]. These features typically must be engineered into proteins, thereby creating a new set of challenges including aggregation and impacts on protein function. Water-in-oil-in-water (W/O/W) nanoparticles are comprised of aqueous pockets that are encapsulated by an oil layer stabilized by surfactants that make the whole particle water-soluble [15]. Surfactants present in the interior layer form reverse micelles (RM), i.e., water-in-oil microemulsions, capable of hydrophilic drug encapsulation [17]. Hydrophilic drugs can be loaded into these systems without high temperatures or pressures that might negatively impact the structural integrity of protein-based drugs. Current W/O/W systems, however, utilize surfactants that tend to denature proteins [18]. Ideal surfactants should be biocompatible, demonstrate high encapsulation efficiencies for a wide variety of biomolecules, and preserve the native structure of encapsulated proteins.

RM systems are widely used in synthetic chemistry and in environmental applications [19]. The most commonly used surfactant system is composed of bis(2-ethylhexyl)sulfosuccinate (AOT). AOT RMs can generally solubilize proteins but fail to preserve their native state [20]. Conversely, a surfactant system composed of 1-decanoyl-rac-glycerol (10MAG) and N,N-dimethyldodecylamine N-oxide (LDAO) has been shown to efficiently encapsulate a wide variety of proteins while preserving their native structures [21]. This mixture is also amenable to complete drying without denaturing the cargo and permits emulsification and encapsulation of cargo without a need for high-shear approaches that can damage macromolecules or promote their aggregation [20,21,22,23,24,25]. In this study, we present modification of the previous 10MAG/LDAO RM platform to optimize biocompatibility while preserving the protein encapsulation advantages of this system.

To date, the 10MAG/LDAO system has been employed in nuclear magnetic resonance (NMR)-based studies for structural and dynamic analysis of proteins and RNA [21]. NMR studies require low-viscosity solvents, such as straight-chain alkanes, to maximize solution NMR performance by promoting rapid tumbling of the RM particle. These solvents, while useful for structural studies, are cytotoxic, thus development of the 10MAG/LDAO system toward biologic delivery requires replacement of the bulk organic solvent with a biocompatible solvent that preserves the general applicability of the system for protein encapsulation. The present study aimed to identify biocompatible organic solvents that facilitate hydrophilic molecule encapsulation and exhibit minimal toxicity. 

A broad array of solvents used in oral and topical drug delivery systems were tested using a small molecule fluorescent dye (propidium iodide, PI) as a model ‘drug’ in an effort to identify most likely biocompatible solvents for further development. This approach allowed us to screen a wide range of formulation conditions without the need for expensive and time-consuming mass production of protein-based cargo. Use of propidium iodide assumes that the behavior of the small molecule cargo will provide predictive value for the encapsulation efficiency of proteins. We validate these predictions by encapsulating a model protein, red fluorescent protein (RFP), to show that this novel screening approach is an efficient and inexpensive way to execute initial optimization of surfactant systems for biological applications.

We used a full factorial screening process that varied surfactant concentration, 10MAG/LDAO molar ratio, and water content of the system to maximize the scope of formulation testing (Appendix A). Our approach also compared two encapsulation methods to evaluate the importance of sample preparation technique versus sample composition. The phase inversion temperature (PIT) [26] approach uses incremental heating to promote emulsification, which may prove detrimental for encapsulation of protein cargo, while the solvent displacement [27] approach utilizes addition of an amphiphilic cosolvent to drive emulsification. Over 200 formulations were assessed for encapsulation efficiency. Each formulation was tested for toxicity against *Saccharomyces cerevisiae* and HeLa cells at three doses. The compositional, encapsulation, and toxicity data were analyzed using a statistical data mining method to identify important correlative relationships. The organic solvent was the dominant compositional factor in determining encapsulation efficiency and toxicity, indicating that other compositional factors (e.g., water content, cosolvent, sample preparation method) are of limited importance in determining formulation performance. Our analysis revealed an important trade-off between toxicity and encapsulation efficiency that correlates with the lipophilicity of the solvent. The predictions offered by the broad PI screen were consistent with test of encapsulation efficiency for RFP.

## 2. Results

### 2.1. Encapsulation Efficiency

Encapsulation efficiency was determined by comparing the fluorescence signal of PI in the aqueous and organic phases after correction for intrinsic fluorescence of the solvent. Figure 1 shows excitation-emission matrix spectra (EEMS) of representative aqueous and organic layers from two formulations. Encapsulation efficiency was evaluated by EEMS for every formulation in this study. The EEMS shows the fluorescence emission intensity as a heat map for a broad range of excitation and emission wavelengths. These data are collected in a grid-like fashion such that the sample is irradiated with a single excitation wavelength for each emission scan. The emission scans at each excitation wavelength are assembled into a matrix for visualization as a contour plot. This means of assessing partitioning of encapsulated fluorescent molecules has the advantage of accounting for spectral shifts that occur due to differences in the bulk solvent dielectric as often occurs for RM-encapsulated hydrophilic fluorophores.

Figure 1 illustrates examples of formulations with poor (iso-octane) and excellent (Capmul) encapsulation efficiency. As shown, iso-octane has no observable fluorescence signal in the organic layer and a strong signal in the aqueous layer (Figure 1a,b), indicating that no observable PI partitioned into the organic phase (Figure 1c,d). Conversely, Capmul MCM shows the opposite result, indicating that most of the PI has partitioned into the organic layer. To assess encapsulation performance comprehensively for formulations prepared in each solvent, partition coefficients were averaged across all formulations prepared in that solvent using either the solvent displacement or PIT approach (Figure 2). Formulations in iso-octane encapsulated approximately five percent of the PI. The scaling of contour values used to generate Figure 1 results in the small PI signal from the iso-octane layer being below the lowest contour. The data collected has a resolution on the order of a single CPS (count per second), thus the sensitivity of the measured data is much greater than shown in the contour plots; this sensitivity is reflected in the calculated encapsulation efficiency values (Appendix A). Formulations prepared in Transcutol and Capmul MCM significantly outperformed all other solvents by encapsulating ~70% of the available PI in the sample, on average. 

### 2.2. Yeast Viability

To evaluate the toxicity of the formulations to eukaryotic cells, *S. cerevisiae* were exposed to each formulation, washed with PBS, and plated on YPD media. Relative growth of yeast colonies was scored to evaluate the toxicity of each formulation (Appendix A). Each formulation was tested at three dilutions, as described in the Materials and Methods (Section 4.4). These toxicity measurements suggested that the solvent used when creating the formulation was a strong determinant of the toxicity of the formulation. As a control, toxicity of each solvent was assessed by treating yeast with samples of solvents containing various concentrations of PI but without 10MAG/LDAO. Yeast treated by these solvent-only control samples were also examined by fluorescence flow cytometry to more thoroughly quantify permeabilization of the cells by the solvents. Positive (cetyltrimethylammonium bromide, CTAB) and negative (phosphate buffered saline, PBS) controls were included in each assay. In these measurements, fluorescence emission of PI is measured on a cell-by-cell basis. PI irreversibly binds to DNA when the integrity of the cell is disrupted, thus this measurement provides a quantitative evaluation of the extent to which the solvents (i.e., without 10MAG/LDAO) permeabilize the cells. It is important to clarify that method distinguished permable cells from non-permable cells. Though permeability is often correlated with cell death, our approach does not directly distinguish live cells from dead cells. Indeed, an ideal cytosolic delivery system should permeabilize the cells without killing them. Here, the permeabilization data acquired by flow cytometry offers insight on permeabilization that are compared to the toxicity measurements (via measurement of yeast viability after treatment). We used these data to evaluate the influence of the solvents, themselves, on the yeast cells more comprehensively. As noted in the discussion, Lauroglycol 90 shows a promising trend of permabilizing cells while maintaining their viability. 

Data for these solvent-only tests, which illustrate the innate toxicity and permeability profiles of the solvents, independent of reverse micelles, are shown in Figure 3. Figure 3a illustrates the flow cytometry data using the controls: PBS with PI (non-permeabilizing) and CTAB with PI (fully permeabilizing). CTAB permeabilized a substantially larger portion of yeast cells compared to PBS, 99.8% and 5.3%, respectively. Each test had approximately that same amount of yeast cells to facilitate comparability of permeabilization values. Four samples were tested for each solvent, one without PI and three with PI at varied dilution (undiluted, diluted 10-fold, and diluted 100-fold). Dilution was used to examine the dose dependence of PI in modulating solvent toxicity. The concentration of PI used in these treatments replicated the PI concentrations used in the formulation dilutions. This dose produces fluorescence signals are near the limit for detection by our flow cytometry instrument. To overcome this limitation, cells were washed with PBS after exposure to the solvent/PI treatments. They were then exposed to a uniform, large dose of PI in aqueous solution to quantify permeabilization. The PI in the aqueous treatment binds to DNA in any cells that were permeabilized by the solvent/PI treatment. This post-treatment staining dominates the flow cytometry response. 

The permeabilization data for the solvent/PI treatments are shown in Figure 3b, where solvents are presented in order from least to most hydrophobic (lowest to highest logP value). The permeabilization of yeast by the solvent/PI samples varies considerably from solvent to solvent and is independent of PI content in the treatment. Figure 3c shows the viability of the yeast cells after the treatment shown in Figure 3b. Transcutol HP, Capmul MCM, and Capryol 90, which have logP values below 3.0, killed the yeast cells. Solvents with logP values greater than 3.0 showed increased cell viability. Figure 3d compares the viability and permeability for each solvent. Generally, higher hydrophobicity of the solvent correlates with higher viability in the treated cells. Solvent toxicity seems loosely correlated to permeability. With the exception of Capmul MCM, permeabilization greater than 70% results in low viability (i.e., high toxicity). Conversely, both Lauroglycol solvents appear to offer a combination of low toxicity and moderate permeabilization of the cells.

### 2.3. HeLa Cell Viability

Toxicity to mammalian cells was assessed using the CellTiter-Blue Assay after exposure of HeLa cells to each solvent or formulation. This assay quantitatively measures conversion of resazurin (blue/purple) to resorufin (pink) via aerobic respiration by live HeLa cells. Resorufin fluorescence after incubation is directly representative of cell viability, thus viable cultures appear pink while cultures with extensive cell death remain purple. This is quantified by absorbance spectroscopy to determine the viability of cells after each treatment compared to controls. PBS treatment provides a negative (non-toxic) control, while treatment with CTAB provides a positive (fully toxic) control. To investigate the toxicity of solvents toward mammalian cells, cultures of HeLa cells were treated with each solvent in the absence of PI and in the presence of PI at three different concentrations (see Section 4.5 in Materials and Methods). The data from these treatments is shown in Figure 4a. The relative toxicity of solvents that were observed for yeast cells were largely preserved in HeLa cells. To evaluate the influence of PI on mammalian cells, solvent-only treatments with and without PI are compared in Figure 4b. The presence of PI does appear to incur a small degree of reduced viability toward mammalian cells. 

HeLa cells were also treated with each formulation at three doses: undiluted, 5-fold diluted, and 25-fold diluted. In each formulation treatment, dilution was performed using the respective organic solvent such that every assay sample received the same total volume of solvent, thus the dose variance corresponds to changing concentration of PI-containing RMs only. Relative viability is shown as violin plots in Figure 4, panels b and c. In these plots, the relative thickness of the violin indicates the percentage of samples within a data set that exhibited the viability shown on the x-axis for each subgroup labeled on the y-axis. Figure 4c shows viability for cells treated with formulations prepared in solvents that were highly toxic to HeLa cells (lethal solvents: Capmul MCM, Capryol 90, and Transcutol HP). Dilution of the RMs did appear to influence toxicity of the formulations, but the dominant influence was caused by solvent. Figure 4d shows viability for cells treated with formulations prepared in non-toxic solvents (iso-octane, Captex 355, Lauroglycol 90, Lauroglycol FCC, and Labrafac PG). Improved viability was evident with dilution of the PI-containing RMs. Across the entire data set, removal of PI has a more moderate effect than dilution of the RMs, thus the apparent mild toxicity of PI does not significantly compromise the value of these data. These data not only indicate promising solvents, but also indicate that RM dose will be an important parameter to optimize in future development.

### 2.4. Correlation Analysis

In total, 24 formulations were prepared in each solvent tested. The screening approach varied total surfactant concentration, LDAO/10MAG ratio, aqueous:organic volume ratio, and sample preparation method for each of eight solvents (Appendix A). The volume of hexanol added to each sample also varied, thus this was included as a compositional variable as well. For each formulation, encapsulation efficiency was determined without dilution, and toxicity towards *S. cerevisiae* and HeLa cells were measured for three dilution conditions, thus every formulation produced six data points corresponding to compositional variables and three data points regarding performance for each dilution. To evaluate this large data set (5164 total data points, Appendix A), R Rattle software was used to statistically analyze pairwise trends in the data via determination of Pearson’s correlation coefficient for each pair of compositional/performance parameters. This analysis randomizes the data such that each pairwise correlation is examined independently of all others. This approach permits an unbiased comparison of the compositional variables and performance metrics assessed in our screen. A global analysis of the data set is presented in Figure 5. Additionally, the data set was divided by preparation method and by solvent to more closely examine trends within each sample set (Appendix A). These trends are briefly described in the Appendix A. 

As shown in Figure 5, some expected correlations are seen, e.g., the negative correlation between surfactant concentration and dilution factor and the positive correlation between HeLa and *S. cerevisiae* toxicity, that provide confidence in the integrity of the analysis. To understand the relationship between formulation composition and performance, other correlations are notable and support the conclusions presented above. The logP value of the solvent correlates negatively with encapsulation efficiency while correlating positively with viability of treated cells. These correlations indicate that higher logP values show the benefit of low toxicity at the cost of strong encapsulation of hydrophilic cargo. Negative correlations are evident between partition coefficient and viability for both HeLa and *S. cerevisiae*, supporting the conclusion that PI, itself, exhibits some level of toxicity in both cell types. This outcome of our study indicates that non-toxic small molecule cargo will be important for future applications of the screening approach presented here. It is also interesting to note that the negative surfactant molarity:viability correlation and the positive dilution factor:viability correlation are stronger for HeLa cells than for *S. cerevisiae*, indicating that the toxicity of the surfactant mixture is stronger for these typically robust mammalian cells. The hexanol used as cosolvent also shows a general trend of mild toxicity suggesting that further development of this system would benefit from optimization of this component. 

In addition to these global trends, the solvent- and method-specific correlation analysis (Appendix A) provide further insight that is more complex. Broadly speaking, the performance of the solvent dominates the observed trends for the ‘lethal’ solvents, as classified in Figure 3. Instructive correlations for the ‘non-lethal’ solvents are addressed in the Appendix A. 

## 3. Discussion

The data presented here represent the first effort to adapt the 10MAG/LDAO system for applications in which biocompatibility is important. Previous uses of this surfactant system have focused on structural studies of proteins and other biopolymers [17,21]. 10MAG/LDAO has proven to be the most versatile system for encapsulating proteins in reverse micelles without disrupting their native fold. In most cases, optimal conditions can be readily identified by a simple screening approach [17]. By replacing the organic solvent with biocompatible solvents that have already been employed in the development of topical, oral, or inhaled drug delivery systems, our primary goal in this study was to identify the key solvent properties that would optimize encapsulation efficiency while keeping toxicity as low as possible. Our analysis revealed a trade-off between efficient encapsulation of hydrophilic cargo and low toxicity. 

We chose to use a small fluorophore, PI, as ‘drug-like’ aqueous cargo to facilitate an inexpensive, broad screen of solvents. A potential weakness of this approach is the assumption that observed trends using a small molecule will be predictive of the system’s performance using protein cargo. Typically, there are two primary challenges to encapsulating proteins in delivery systems. Most delivery vehicles disrupt the structural integrity of the protein cargo by non-specific interactions with the encapsulation matrix or with surfactants/lipids employed [6,16,28]. The 10MAG/LDAO system tends to minimize such unfavorable interactions. Delivery vehicles that do not tend to disrupt the protein structure, e.g., liposomes, suffer from poor encapsulation efficiency, thereby necessitating the production of very large quantities of protein, much of which fails to encapsulate [6,28]. Our tests with PI employed two typical formulation preparation methods that both utilize an excess aqueous phase, thereby potentially leading to similar waste of non-encapsulated cargo. The direct-injection, or self-nanoemulsion, method is an alternative approach for RM encapsulation that employs a very small volume of aqueous solution that is entirely encapsulated when conditions are optimized. This method has been used for application of the 10MAG/LDAO system for structural studies of proteins with great success [17], thus we applied this approach using the solvents tested in our PI screen to examine the validity of the observed relationship between logP and encapsulation efficiency revealed by the small molecule cargo.

The red fluorescent protein (RFP) mCherry [29] was injected into each representative solvents using an established screening approach [17]. Briefly, this method involves injection of several microliters of a high-concentration protein solution into a mixture of 10MAG/LDAO in organic solvent followed by titration with hexanol to optimize encapsulation (Appendix A). Optimal conditions are identified by clarification of the sample’s visual appearance (i.e., the sample transitions from cloudy to clear by eye). Two important advantages of this approach are the elimination of a bulk aqueous phase, thereby minimizing wasted protein cargo, and the minimization of hexanol in the sample, which showed mild toxicity to both types of cells tested in our PI-based screen. We tested encapsulation of RFP in iso-octane, Capmul MCM, and Lauroglycol 90 to examine the predictive value of our PI screen for encapsulation of proteins. Iso-octane, an alkane, is frequently used in RM systems, and has been used in the past for encapsulation of proteins for structural studies [30,31,32]. Capmul MCM showed high encapsulation efficiency with PI, low permeabilization of yeast cells, and high toxicity to both yeast and HeLa cells, thus it represents the ‘lethal’ solvent for this test. Lauroglycol 90, conversely, showed moderate encapsulation efficiency with PI, high permeabilization of yeast cells, and low toxicity to both yeast and HeLa cells, thus it was selected as representative of the best-performing ‘non-lethal’ solvent. 

EEMS of the optimal RFP encapsulation samples for each of these solvents are presented in Figure 6. The PI-based screen revealed a negative correlation between encapsulation efficiency and the hydrophobicity of the bulk solvent, as represented by the solvent LogP value. Capmul MCM, among the least hydrophobic solvents tested (LogP 2.27), encapsulated a majority of the protein injected, yielding an efficiency of 61.0%. This favorable protein encapsulation performance mirrors the strong encapsulation of PI in the large-scale screen. RFP encapsulation in iso-octane (LogP 3.08) was highly efficient, encapsulating 68.5% of the RFP injected into the sample. This result is consistent with the known performance of the alkanes in facilitating high encapsulation of proteins in the 10MAG/LDAO system. Lauroglycol 90 (LogP 3.83) was among the most hydrophobic solvents tested in the PI-based screen. As was seen for PI, Lauroglycol 90 showed the lowest RFP encapsulation efficiency with 11.6% of the protein encapsulated. Despite this reduced performance compared to the less hydrophobic solvents, this encapsulation efficiency is comparable to that of protein-compatible systems such as liposomes. The self-nanoemulsion RM encapsulation approach, however, avoids the need for large bulk aqueous phases that create significant waste of non-encapsulated protein cargo in liposome preparations. It also avoids harsh preparation methods needed for liposome-based encapsulation (e.g., freeze–thawing, extrusion) that negatively impact protein stability [28,33].

The broad screen using PI as cargo showed that the predominant factor in determining toxicity of formulations was the bulk organic solvent used. The protein encapsulation tests reproduce the encapsulation efficiency trends seen using the small molecule cargo. This correlation suggests that the use of inexpensive, small molecule, hydrophilic cargo for broad screening provides strong predictive value for performance of the 10MAG/LDAO system for use with protein-based cargo. The findings presented here lay the foundation for further development of this system toward both long-term storage of therapeutic protein cargo and incorporation into W/O/W or SLNP systems for delivery of protein-based therapeutics [16,28]. 

Protein-containing samples prepared using the 10MAG/LDAO system in alkane solvents are often stable for weeks to months at room temperature. The RFP-containing samples prepared in iso-octane showed consistent fluorescence emission over the span of weeks when stored at room temperature (10% loss over two weeks). Long-term storage of protein therapeutics remains a significant challenge, generally necessitating expensive infrastructure (i.e., freezers) that only moderately prolongs shelf life. Development of a system in which proteins can be stored at room temperature for long periods, then recovered and transitioned to delivery systems, could be an attractive new approach for mitigating this challenge. The high encapsulation efficiency offered by the most hydrophobic solvents tested here present the potential for development of such an approach using the 10MAG/LDAO system. Optimization of this approach is an ongoing avenue of inquiry in our group. 

Hybrid nanoparticle systems (e.g., W/O/W and SLNPs) for delivery of hydrophilic small molecule drugs are quite mature [13,34,35]. Incorporation of the 10MAG/LDAO mixture into such approaches may facilitate broad application of these methods for delivery of protein-based therapeutics. The results presented here inform such further development. The range of logP values tested here identifies an optimal range (LogP ~3.5) for future screening efforts using solvents that work well in W/O/W or SLNP systems and subsequent tests of drug delivery performance via studies of cargo uptake. Indeed, previous studies have demonstrated the utility of Lauroglycol 90 as a primary component in W/O/W microemulsions for encapsulation of hydrophilic cargo [36]. Lauroglycol 90 was the best-performing solvent in the present study in terms of balancing moderate encapsulation efficiency against low toxicity, thus future efforts to incorporate the 10MAG/LDAO system into hybrid nanoparticle systems will focus on solvents with similar hydrophilicity. Overall, the data presented here suggest long-term potential for the 10MAG/LDAO mixture to find application in storage or delivery of protein-based therapeutics. 

## 4. Materials and Methods

### 4.1. Chemicals and Cell Lines

The solvents (logP) used were: Transcutol HP (0.03180) (Gattefosse, Paramus, NJ, USA), Capmul MCM (2.27090) (Abitec, Janesville, WI, USA), Capryol 90 (3.00) (Gattefosse, Paramus, NJ, USA), iso-octane (3.078600) (Avantor Performance Materials LLC, Radnor, PA, USA), Captex 355 (3.26680) (Abitec, Janesville, WI, USA), Lauroglycol 90 (3.83130) (Gattefosse, Paramus, NJ, USA), Lauroglycol FCC (3.83130) (Gattefosse, Paramus, NJ, USA), and Labrafac PG (8.44270) (Gattefosse, Paramus, NJ, USA). The logP value is a numerical representation of lipophilicity, the ratio of the concentration when partitioning between an oil and lipid phase [37]. Surfactants used to make the reverse micelles were N,N-dimethyl-1-dodecylamine N-oxide (LDAO) (BeanTown Chemical, Hudson, NH, USA), and 1-decanoyl-*rac*-glycerol (10MAG) (TCI America, Portland, OR, USA). Phosphate buffered saline (PBS) (250 mM sodium chloride, 50 mM sodium phosphate, Alfa Aesar, Ward Hill, MA, USA) was used as the water loading phase.

Propidium iodide (MP Biomedicals, LLC., Solon, OH, USA), a water-soluble, fluorescent, drug-like molecule, was used as cargo formulations prepared using solvent displacement and phase-inversion temperature techniques. 1-Hexanol (Alfa Aesar, Ward Hill, MA, USA) was used as a co-solvent when appropriate (see Appendix A). 

Yeast extract–peptone–dextrose media was prepared from yeast extract (BD Biosciences, San Jose, CA, USA), bacto peptone (BD Biosciences, San Jose, CA, USA), and dextrose (Sigma-Aldrich, St. Louis, MO, USA). *S. Cerevisiae* strain BY4741 (MATa his3Δ1 leu2Δ0 met15Δ0 ura3Δ0) and HeLa cells (gift from the laboratory of Dawn Carone at Swarthmore University) were used to assess toxicity against eukaryotic cells. Cetrimonium bromide (CTAB) (VWR, Radnor, PA, USA) and phosphate buffered saline (Sigma-Aldrich, St. Louis, MO, USA) were used as positive and negative controls, respectively, in cellular assays. Toxicity in HeLa cells was assessed using the CellTiter-Blue Cell Viability Assay (Promega, Madison, WI, USA) after culturing in Dulbecco’s Modified Eagle’s Medium (DMEM) cell culture media (VWR, Radnor, PA, USA). 

mCherry red fluorescent protein (RFP) was prepared by recombinant expression in BL-21 CodonPlus (DE3) RIL *E. coli* cells (Agilent Technologies, Santa Clara, CA, USA). Protein was purified by two-stage salt cut (50% and 65% ammonium chloride) followed by column purification on Q-sepharose (Cytiva, Marlboro, MA, USA). Pure RFP was concentrated and exchanged to PBS prior to encapsulation. 

### 4.2. Sample Preparation

A factorial screening approach was used to comprehensively sample the compositional space of formulations. This approach separately varied surfactant concentrations, 10MAG/LDAO ratios, and volume ratio of organic-to-aqueous solvents. An example of the sample compositions tested for each solvent can be found in the Appendix A. Sample composition was optimized for maximal encapsulation efficiency in a step-wise fashion. Surfactant molarity was first varied while holding the 10MAG:LDAO (M/M) ratio and PBS:solvent ratio (*v*/*v*) constant. The surfactant molarity generating the greatest encapsulation efficiency was used for the next round of optimization in which 10MAG:LDAO ratio was varied using a constant PBS:solvent ratio. The final round optimized the PBS:solvent ratio using optimal surfactant molarity and 10MAG:LDAO ratios. In this fashion, conditions for maximal encapsulation efficiency were identified for each solvent. 

The screening approach was performed using two methods, the solvent displacement approach and the phase inversion temperature (PIT) method, for each solvent. Solvent displacement samples [27] were made by adding the corresponding amounts of 10MAG, LDAO, solvent, PI stock solution (686 µM in PBS), and PBS to a screw-cap glass vial with a PTFE-lined (polytetrafluoroethylene) lid. Vials were sealed and wrapped with PTFE tape to prevent evaporation, then mixed by vortexing until surfactants fully dissolved. If phase separation did not occur, hexanol was titrated as a cosolvent in 10 µL steps with ten-minute settling time after each addition until separation occurred. For the PIT method, samples were prepared in an identical fashion to solvent displacement except that, instead of adding cosolvent to promote phase separation, the samples were heated in a step-wise fashion from 40 °C to 90 °C [26]. This heating process was executed using 10 °C intervals in which the sample was heated for ten minutes, vortexed for three minutes, then permitted to settle at room temperature for ten minutes. If no separation occurred after heating the sample at 90 °C, then hexanol was added by titration as described above.

To determine the toxicity of the samples, the organic phase was removed from each formulation after measurement of fluorescence spectra. The undiluted organic phase was tested without further alteration. A small volume of the undiluted organic phase was diluted in the respective solvent to 5-fold and 25-fold dilutions; these dilutions were also tested for toxicity. Additional control experiments were performed to determine the toxicity of the solvents and the potential for PI to partition into the solvent without 10MAG:LDAO present. Four samples were made for each organic solvent, one of pure solvent and three with varying concentrations of injected PI solution (18.5 µM. 0.185 µM, and 0.0185 µM). For some solvents, all injected PI was dispersed evenly throughout the sample, while for others, the PI solution settled into a distinct phase. The organic portion of each sample was tested to examine toxicity as described below.

To determine encapsulation efficiency for proteins, a subset of formulations was tested for encapsulation of RFP via the self-nanoemulsion, also referred to as direct injection, method with titration of hexanol to optimize encapsulation efficiency as described in detail elsewhere [17]. The subset of formulations tested for protein encapsulation were selected based on formulation performance using PI. Capmul MCM, iso-octane, and Lauroglycol 90 were tested using the optimal surfactant ratios and molarities from the PI screen (see Appendix A) and water-loadings (*W*_0_, water:surfactant molar ratio) of 15 and 20 using a 39.7 µM solution of RFP in PBS.

### 4.3. Encapsulation Efficiency

Excitation-emission matrix spectra (EEMS) were collected of the aqueous and organic phases of each formulation to determine the efficiency of PI encapsulation. EEMS are created by collecting emission spectra at a single excitation wavelength; then the excitation wavelength is iterated to the next increment to collect another emission spectrum. This process is repeated for a range of excitation wavelengths to create a matrix of emission intensity values for each excitation and emission wavelength sampled, thereby fully characterizing the emission character of the fluorescent species. EEMS were collected on a Fluoromax-4 (Horiba Scientific, Piscataway, NJ, USA) using 10 nm steps for excitation wavelengths from 450 nm to 580 nm and emission wavelengths from 600 nm to 700 nm, 0.1000 s integration time, and 5 nm excitation and emission slits for all PI-containing formulations. For protein-containing samples, EEMS were collected using 10 nm increments for excitation wavelengths from 500 nm to 630 nm and emission wavelengths from 580 nm to 680 nm, 0.1000 s integration time, and 2 nm excitation and emission slits.

Encapsulation efficiency for PI-containing samples was determined from the EEMS data as follows. First, as shown in Equation (1), the maximum peak emission intensity was determined from the EEMS of the organic phase for each formulation (*O_F_*). EEMS were collected for each organic solvent tested. Intrinsic fluorescence in the organic solvent (*O_I_*) at the excitation-emission wavelength combination corresponding to that of *O_F_* was subtracted to determine a corrected fluorescence intensity for each formulation organic phase (*O_C_*).
*O_F_* − *O_I_* = *O_C_*(1)

As shown in Equation (2), the corrected fluorescence was used to calculate the encapsulation efficiency, *E_e_*.
(2)$$\frac{O_{C}}{A+O_{C}}\times 100=E_{e}$$
where *A* represents the PI emission intensity as determined from the EEMS of the aqueous phase for each formulation. *E_e_* represents an estimate of the percentage of PI fluorescence observed in the organic phase for each formulation. While this approach does not correspond strictly to a partition coefficient due to solvent-dependent changes in emission intensity, it provides a uniform approach for evaluating the relative encapsulation efficiency of each formulation.

For protein encapsulation samples, encapsulation efficiency was assessed by comparing the RFP fluorescence intensity in self-nanoemulsion samples to that of an aqueous solution of RFP representing a condition of 100% encapsulation. For example, a self-nanoemulsion sample prepared using the 39.7 µM RFP stock solution at a *W_0_* of 15 in 75 mM surfactant would result in an RFP concentration of 806 nM if all RFP encapsulated successfully, thus the aqueous RFP concentration used for encapsulation efficiency calculation would be 806 nM for this sample. As shown in Equation (3), the intensity of RFP in the organic, *O_RM_*, was corrected for intrinsic emission of the solvent, *O*, at the excitation and emission wavelengths of maximum RFP emission. This difference corresponded to emission from encapsulated RFP. This was compared to the maximum emission intensity from the aqueous RFP sample, *A_RFP_*, to calculate a percentage of total RFP encapsulated.
(3)$$\frac{\left(O_{RM}-O\right)}{A_{RFP}}\times 100=E_{e}$$

### 4.4. S. cerevisiae Viability and Permeability

To determine yeast cell permeabilization and viability, *S. cerevis*iae (strain BY4741) was evaluated using flow cytometry and subsequent growth assays, respectively. To evaluate permeabilization, yeast was grown overnight in 5 mL of standard YPD media to saturation. The following morning, 0.5 mL saturated yeast culture was resuspended in 4.5 mL of fresh YPD media and grown for 4–6 h before transferring 100 µL of yeast to a new microcentrifuge tube. Microcentrifuge tubes with yeast were then spun down at 4000 rpm for 4 min, the supernatant was removed, and 400 µL of the solvent/PI sample was added to the tubes for 10 min at room temperature. Solvents were tested without PI and with PI at undiluted, 10-fold diluted, and 100-fold diluted conditions (see Appendix A for details). Positive and negative controls for permeabilization were 0.1% CTAB or PBS plus 5 µg/mL PI, respectively. Following treatment, cells were spun down again at 4000 rpm for 4 min. The supernatant was removed, pellets were then washed once with 1000 µL of PBS, pelleted again, and finally resuspended in 500 µL of PBS plus 5 µg/mL PI and incubated for 10 min before evaluation by flow cytometry. Incorporation of PI dye into yeast was measured on a BD FACSCelesta flow cytometer in a 96-well round bottom plate using 488 nm laser excitation and a 575 nm emission filter. A total of 10,000 cells per sample were counted and the percentage of cells found to be permeable for all samples was established by gating around cells on a histogram illustrating PI signal with the positive control, the known permeabilization reagent, 0.1% CTAB.

To test for viability after the permeabilization assay, 5 µL of the yeast sample was removed from the 96-well plate and grown on a YPD plate at 30 °C overnight. After the samples were incubated for 48 h, plates were photographed and qualitatively scored for high, low, or no growth.

### 4.5. HeLa Cell Viability

HeLa cells were treated with RM formulations to evaluate cytotoxicity in a model mammalian cell line. Cells were seeded in a 96-welled plate at 100,000 cells per well, grown for 24 h in 180 µL of DMEM, 10% fetal bovine serum, 1% penicillin/streptomycin, 1% L-glutamine in a standard CO_2_ incubator with 5% CO_2_ at 37 °C. Cultures were exposed to RM formulations by addition of 10% volume of the formulation organic phase for an additional period of 24 h. Each formulation was tested undiluted, at 5-fold dilution, and at 25-fold dilution with dilutions prepared using the pure organic solvent corresponding to that used for original formulation preparation. As a positive control for cytotoxicity, CTAB reagent was used at final concentration of 0.3% (completely cytotoxic). Addition of 20 µL of PBS was used as a negative control. Subsequently, cytotoxicity was assayed in HeLa cells using CellTiter-Blue reagent. In this assay, active cellular metabolism is evaluated by monitoring the enzymatic conversion of resazurin to resorufin which exhibits bright fluorescence emission at 590 nm. Plates were incubated at 37 °C for 2 h with addition of 20 µL of CellTiter-Blue reagent. Fluorescence emission intensity at 590 nm was measured on a Synergy HT plate reader using an excitation wavelength of 485 nm. Measured fluorescence emission intensity is directly representative of HeLa cell viability after treatment.

### 4.6. Data Analysis

Correlation analyses were performed using R rattle software. A Pearson’s correlation calculation (Equation (4)) was used to determine the strength in relationships between all variables for each formulation.
(4)$$r=\frac{n\left(\sum xy\right)-\left(\sum x\right)\left(\sum y\right)}{\sqrt{\left(n\sum X^{2}-{\left(\sum x\right)}^{2}\right)\left(n\sum y^{2}-{\left(\sum y\right)}^{2}\right)}}$$

For Equation (4), representing a comparison of two variables *x* and *y*, *n* is the number of observations, *x* is the value of variable *x*, *y* is the value of variable *y*, *r* is the Pearson’s correlation coefficient. Correlation plots were built from these calculated values to examine the relationships between variables for the entire data set, as well as separately to examine the strength in relationships specific to each organic and preparation method.

## 5. Conclusions

This study presents a novel approach for early-stage development of a biocompatible system for encapsulating hydrophilic cargo, especially proteins. As large-scale production of proteins is expensive and often challenging, we employed a hydrophilic small molecule to test a wide range of compositions and to identify important relationships between the solvent used and the performance of the formulation. Our analysis revealed a trade-off between the encapsulation efficiency and toxicity of formulations that depends on hydrophobicity of the solvent. Lauroglycol 90 (logP 3.83) offered the most optimal balance between these attributes for the small molecule cargo. We encapsulated RFP in three solvents to test the predictive value of the small molecule screen and found that the encapsulation efficiency of RFP scaled similarly with hydrophobicity of the solvent. This study offers the foundation for future development of the 10MAG/LDAO system toward applications in which biocompatibility is critical such as storage of protein-based therapeutics and W/O/W or SNLP-based drug delivery systems.

## Figures and Tables

**Figure 1 molecules-27-01572-f001:**
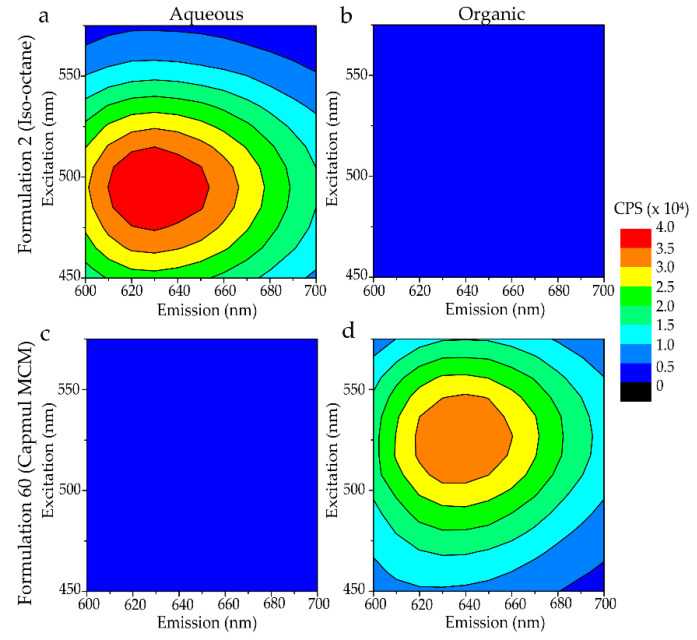
Example EEMS data are illustrated as contour plots using a heat map coloring scheme as indicated. Here, the numerical value of the contour represents the emitted fluorescence intensity in counts per second (CPS). Spectra are shown for the aqueous (**a**,**c**) and organic (**b**,**d**) layers of Formulations 2 (**a**,**b**) and 60 (**c**,**d**) which were prepared in iso-octane and Capmul MCM, respectively. High intensity in the organic layer indicates high encapsulation of PI. High intensity in the aqueous layer corresponds to poor encapsulation efficiency. Details on formulations, listed by numerical index, are provided in Appendix A.

**Figure 2 molecules-27-01572-f002:**
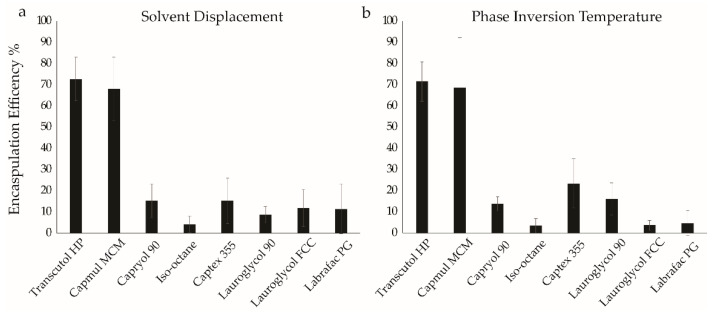
The average encapsulation efficiency is shown, by solvent, for all formulations prepared using the solvent displacement method (**a**) or the PIT method (**b**). Each bar represents the average of 12 samples, and the error bars represent the uncertainty of the mean. Solvents are listed in order from least lipophilic (Transcutol HP) to most lipophilic (Labrafac PG).

**Figure 3 molecules-27-01572-f003:**
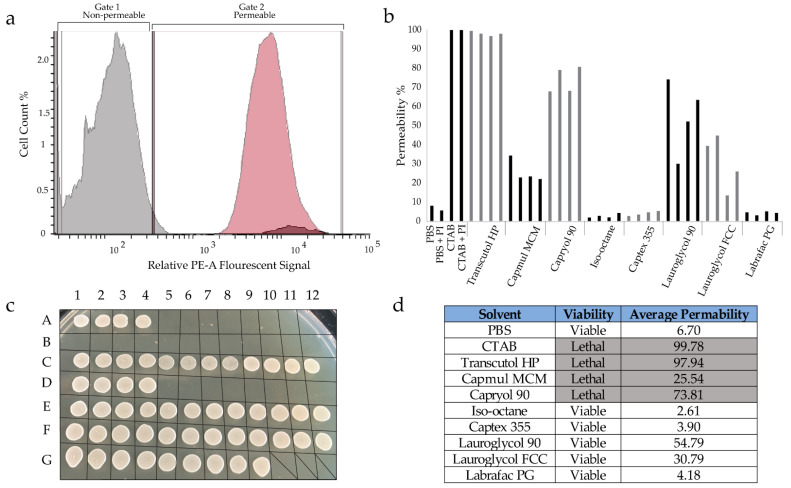
(**a**) Flow cytometry output is shown for control treatments with PBS (gray, negative control) and CTAB (red, positive control). The CTAB treatment exhibits a large number of cells with strong fluorescence in the red (PE-A channel), indicating permeabilization of these cells. Treatment with PBS yields high cell counts with low fluorescence, indicating minimal permeabilization of the cells to PI. (**b**) The permeabilization of *S. cerevisiae* after treatment with solvents containing varying concentrations of PI are shown, as quantitated by flow cytometry. Relative permeabilization was calculated for each treatment using CTAB treatment as representative of 100% permeabilization. Four PI contents were tested for each solvent (presented from lowest to highest logP value). (**c**) The viability of yeast cells after plating on YPD media is shown for all conditions tested. The controls, spotted in duplicate, are shown in positions A1-A8 as follows: PBS with (A1-2) and without PI (A3-4), CTAB with (A5-6) and without PI (A7-8). Solvent-only treatments were spotted in duplicate for four conditions each, in the following order: without PI, with PI undiluted, with PI diluted 10-fold, with PI diluted 100-fold. Thus, eight culture spots are shown for each solvent in order as follows: Capmul MCM (A9-B4), Capryol 90 (B5-B12), Lauroglycol 90 (C1-C8), Lauroglycol FCC (C9-D4), Transcutol HP (D5-D12), Labrafac PG (E1-E8), iso-octane (E9-F4), Captex 355 (F5-F12). Colonies in row G are additional control treatments of buffer without PI (G1-2) and with PI varying doses: undiluted (G3-4), 10-fold diluted (G5-6), 100-fold diluted (G7-8) also spotted in duplicate. (**d**) This table summarizes the relative toxicity of solvents of yeast cells from panel C and an average permeabilization value for each solvent from the data in panel B.

**Figure 4 molecules-27-01572-f004:**
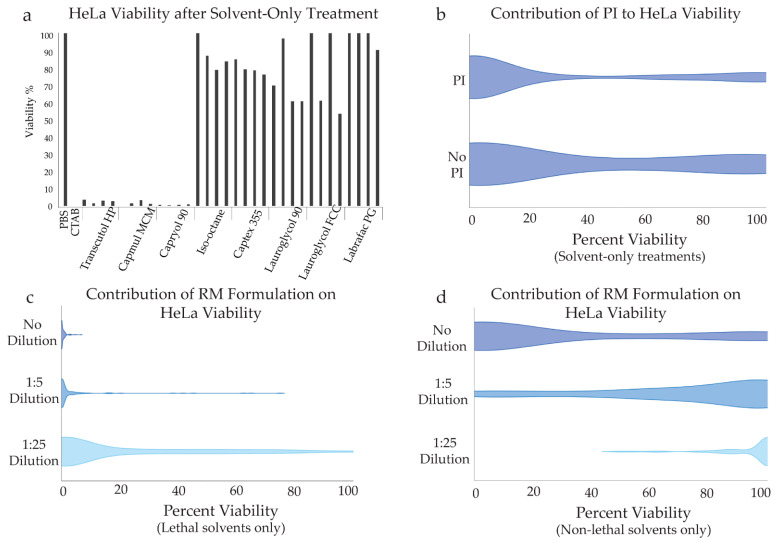
(**a**) HeLa cell viability measurements from the CellTiter Blue assay for PBS and CTAB controls and for treatments of solvent-only without PI, and with PI undiluted, 10-fold dilute, and 100-fold diluted, respectively for each solvent, as labeled. (**b**) A violin plot is shown for all solvent-only treatments, comparing treatment with PI versus treatment without PI. Thickness of the plot indicates relative percentage of samples with viability as indicated on the x-axis within the sample set that received the treatment indicated on the y-axis. (**c**) HeLa cell viability is shown for all lethal solvents (solvents with minimal viability in panel a), illustrating slight reduction in toxicity due to dilution of RMs. (**d**) HeLa cell for all non-lethal solvents (solvents with high viability in panel a) showing that strong viability is seen at the lowest RM concentrations tested.

**Figure 5 molecules-27-01572-f005:**
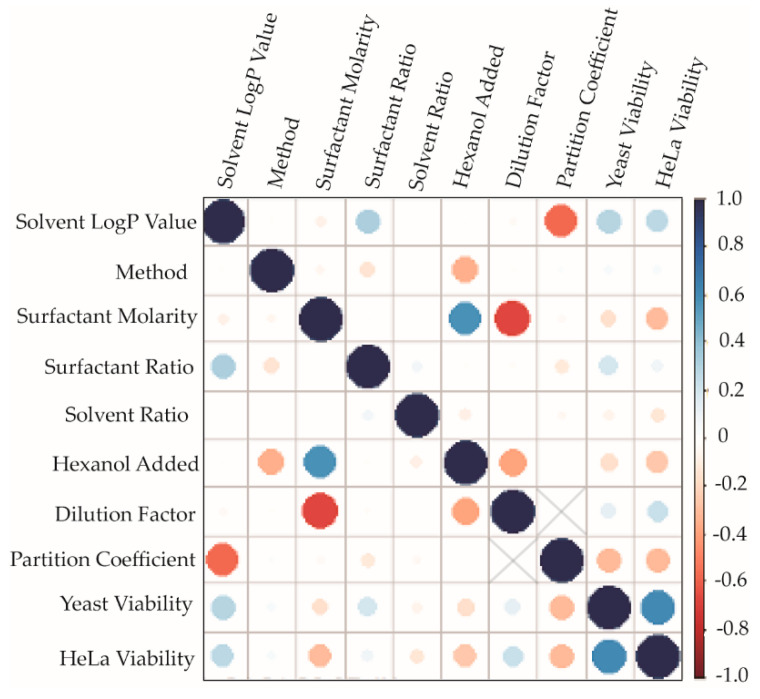
The Pearson’s correlation is shown for each pairwise set of compositional variables and performance measurements. Circle size scales with strength of the correlation. The correlation coefficient values are color coded, as indicated on the right. Dilution factor cannot be compared with partition coefficient, thus correlation data are not provided for this pair.

**Figure 6 molecules-27-01572-f006:**
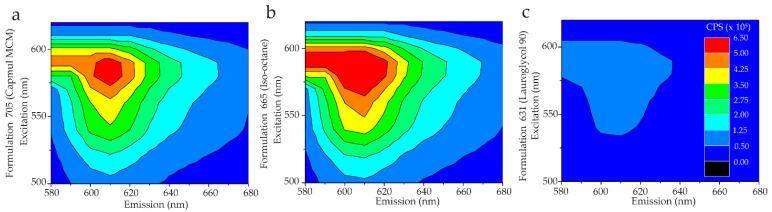
EEMS of RFP are shown as contour plots for self-nanoemulsion formulations prepared in (**a**) Capmul MCM, (**b**) iso-octane, and (**c**) Lauroglycol 90. Contours represent emission intensity in counts per second as indicated by the color bar in panel (**c**). Encapsulation efficiencies were calculated to be 61.0%, 68.5%, and 11.6%, respectively.

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
