# Peer review of "Optimization of Biocompatibility for a Hydrophilic Biological Molecule Encapsulation System"

_molecules, 2022, doi:10.3390/molecules27051572_

Round 1
Reviewer 1 Report
The authors present a study about the encapsulation efficiency and toxicity of the 10MAG/LDAO system by varying the solvents used for the loading of the reverse micelle system. Despite the high amount on data that are generated in the study, there are some major concerns that needed to be addressed.
1) What is the novelty of the study? Please make it more clear in the abstract and the introduction?
2) The encapsulation efficiency of the propidium iodide and the RFP was compared. Why not the biocompatibility?
3) For the toxicity studies of the yeast cells, solvent plus PI was used, but not for the HeLa cells. Why?
4) The reverse micelle system should be loaded with proteins and used for drug delivery. Why are there no toxicity studies on the reverse micelle system? And which influence have the different solvents on the toxicity of the actual drug delivery system? They are loaded with a fluorescent dye or RFP. There should be uptake studies in yeast or HeLa cells possible to show if this system is good for drug delivery.
Author Response
We thank the reviewer for thorough evaluation and feedback. We address the reviewer's concerns point by point below. All edits to the manuscript in response to reviews are shown in the revised manuscript by red text.
1) What is the novelty of the study? Please make it more clear in the abstract and the introduction?
Response to reviewer: We have clarified the novelty of the study in the abstract, introduction, and in the newly-added concluding paragraph. We have also clarified the nature of our study by changing the title of the paper to better reflect the main focus of the study.
2) The encapsulation efficiency of the propidium iodide and the RFP was compared. Why not the biocompatibility?
Response to reviewer: We thank the reviewer for this important question. RFP must be manufactured by recombinant purification from bacterial culture. Though it is a relatively high-yield preparation (yielding 30-40 mg RFP/liter of cells in our hands), producing sufficient quantities of RFP to screen conditions as broadly as the very inexpensive propidium iodide would require purification from hundreds of liters of cells. Additionally, many previous studies have demonstrated that RFP is not harmful to yeast or to HeLa cells, so it is unlikely that the presence of RFP would alter the biocompatibility of our mixtures. Our view, and a strength of our approach, is that the biocompatibility data presented for propidium iodide are broadly representative of the behavior of the 10MAG/LDAO mixtures. As we mentioned in the discussion, the data presented here offer a foundation for further development of the delivery system using solvents that offer a balance of strong encapsulation efficiency and low toxicity. The establishment of an efficient method for such early stage optimization is a key part of this study’s impact. We agree that these points should have been made more clearly. We have clarified these points in the introduction and discussion.
3) For the toxicity studies of the yeast cells, solvent plus PI was used, but not for the HeLa cells. Why?
Response to reviewer: We apologize for the confusion on this point. The tests of solvent only and solvent with PI on HeLa cells were done. We recognize that the descriptions of these experiments and the distinction between tests carried out with solvent-only versus those with RMs (formulations) was poor. We also neglected to include the solvent-only data in the Supplementary Materials. We have edited the manuscript heavily to improve this distinction, and we have revised Figure 4 and the description of the HeLa studies, and added a table (Table S6) to the Supplementary Materials to report the data for these treatments.
4) The reverse micelle system should be loaded with proteins and used for drug delivery. Why are there no toxicity studies on the reverse micelle system? And which influence have the different solvents on the toxicity of the actual drug delivery system? They are loaded with a fluorescent dye or RFP. There should be uptake studies in yeast or HeLa cells possible to show if this system is good for drug delivery.
Response to reviewer: We appreciate the reviewer’s careful evaluation of our work. Our study includes toxicity information for every formulation prepared with propidium iodide against both yeast and HeLa cells. As noted above, the manuscript has been edited to better clarify which data correspond to formulations versus those carried out with solvent alone. The formulation data are reported in concise format in Figure 5 where we present a comprehensive comparative analysis of the wide range of compositional variables tested versus the performance of the formulations, including toxicity information for three dilutions per formulation. These data are analyzed for each solvent in Supplemental Figure 1, and the raw data are presented in the Supplemental Tables. We did not broadly screen RFP-containing formulations for the reasons given in our response to question 2.
The reviewer also asks about testing of the ‘actual drug delivery system’. We recognize that our original manuscript was unclear in distinguishing tests of samples that include RMs versus those that included solvent alone or solvent with PI. We have heavily edited the manuscript and figures to better clarify these points. Importantly, however, we specify in the discussion that our study is a foundational one aimed toward the future development of a true delivery system or a long-term storage platform for proteins. The present study aims to show the promise of this system for encapsulating proteins for biomedical applications. We have edited the description of the study to more accurately reflect the impact of our work and to clarify that we are not yet at the stage of testing appropriateness for true drug delivery.
The reviewer also asks about uptake studies. We feel that uptake studies are appropriate follow-up work for this system after further development, especially after incorporation of the optimal solvent/10MAG/LDAO system in a water-in-oil-in-water microemulsion. We did attempt to investigate uptake in these initial studies, however. Every formulation was tested via flow cytometry to evaluate cell-association/permabilization of yeast cells. The flow cytometer cannot, however, distinguish between live cells or dead cells, and propidium iodide binds irreversibly to DNA, thus these data cannot tell us whether the cells that appear to have taken up the PI are or are not still alive. We chose not to report these data due to this important ambiguity. In fact, it was these data that initially indicated the strong relationship between the solvent and the toxicity of the formulation. The solvent is, by far, the dominant determinant of biocompatibility in our systems. This is why we reported only the solvent-based tests of permeabilization. We have added clarifications of this in the manuscript.
With respect to investigating uptake of RFP by the HeLa cells, we did investigate this via fluorescence microscopy, and the results were also ambiguous. Though some samples showed evidence of uptake, the response was not reproducible enough to responsibly report. We believe this variability in behavior was primarily due to the fact that none of our formulations are water-soluble in their current form, thus mixing the cells with the formulations to homogeneity is not really possible. We fully intend to repeat such experiments with W/O/W mixtures (water-soluble versions of these formulations) as we develop this system further.
Reviewer 2 Report
The manuscript describes an interesting study of an effective screening method for the optimization of a delivery system of hydrophilic proteins. The formulations are based on 10MAG/LDAO creating WOW reverse micelles with different cosolvents, settings, and ratios. Propidium iodide was used as a model compound instead of a protein drug. The efficiency of encapsulation and toxicity were determined to evaluate the best condition.
Major points:
No
Minor points:
1) Font and size in figures are not well readable (figure 1 and 2).
2) A short conclusion with highlighted results can improve the impact.
3) Table S2: (hexanol and HeLa Viability) should be rounded to have a reasonable number of decimal places.
4) References 35, 36, 37, and 38 have a different format of the name than other references.
5) line 573: 0.3 instead of 0.3 will be more consistent with the rest of the numbers
Author Response
We thank the reviewer for these constructive comments. We have increased the font size in Figures 1 and 2, added a short Conclusion section, and made the minor edits to Table S2, the references, and line 573, as pointed out by the reviewer
Round 2
Reviewer 1 Report
The authors carefully revised their manuscript and managed to clarified their work and the intention of the study. All my questions were answered in detail. Therefore I recommend this manuscript for publication.
Author Response
We thank the reviewer for the helpful feedback and are very happy with the improvements to the manuscript.